# Comparison of Different Vascular Biomarkers for Predicting In-Hospital Mortality in Severe SARS-CoV-2 Infection

**DOI:** 10.3390/microorganisms12010229

**Published:** 2024-01-22

**Authors:** Renáta Sütő, Marianna Pócsi, Miklós Fagyas, Edit Kalina, Zsolt Fejes, Zoltán Szentkereszty, János Kappelmayer, Béla Nagy Jr.

**Affiliations:** 1Department of Laboratory Medicine, Faculty of Medicine, University of Debrecen, 4032 Debrecen, Hungary; sutorenata@gmail.com (R.S.); pmarcsi89@gmail.com (M.P.); ekalina@med.unideb.hu (E.K.); fejes.zsolt@med.unideb.hu (Z.F.); kappelmayer@med.unideb.hu (J.K.); 2Doctoral School of Kalman Laki, Faculty of Medicine, University of Debrecen, 4032 Debrecen, Hungary; 3Gyula Kenézy Campus, Intensive Care Unit, Faculty of Medicine, University of Debrecen, 4032 Debrecen, Hungary; dr.szentkereszty.zoltan@kenezy.unideb.hu; 4Division of Clinical Physiology, Department of Cardiology, Faculty of Medicine, University of Debrecen, 4032 Debrecen, Hungary; fagyasmiklos@med.unideb.hu

**Keywords:** SARS-CoV-2, intravascular inflammation, COVID-19 disease, endothelial dysfunction, ACE2, platelets, biomarker, outcome

## Abstract

Severe SARS-CoV-2 elicits a hyper-inflammatory response that results in intravascular inflammation with endothelial injury, which contributes to increased mortality in COVID-19. To predict the outcome of severe SARS-CoV-2 infection, we analyzed the baseline level of different biomarkers of vascular disorders in COVID-19 subjects upon intensive care unit (ICU) admission and prior to any vaccination. A total of 70 severe COVID-19 patients (37 survivors and 33 non-survivors) were included with 16 age- and sex-matched controls. Vascular dysfunction was monitored via soluble VCAM-1, E-selectin, ACE2 and Lp-PLA2, while abnormal platelet activation was evaluated by soluble P-selectin and CD40L in parallel. These results were correlated with routine laboratory parameters and disease outcomes. Among these parameters, VCAM-1 and ACE2 showed significantly higher serum levels in COVID-19 patients with early death vs. convalescent subjects. VCAM-1 was significantly correlated with the Horowitz index (r = 0.3115) and IL-6 (r = 0.4599), while ACE2 was related to E-selectin (r = 0.4143) and CD40L (r = 0.2948). Lp-PLA2 was altered in none of these COVID-19 subcohorts and showed no relationship with the other parameters. Finally, the pre-treatment level of VCAM-1 (≥1420 ng/mL) and ACE2 activity (≥45.2 μU/mL) predicted a larger risk for mortality (Log-Rank *p* = 0.0031 and *p* = 0.0117, respectively). Vascular dysfunction with endothelial cell activation is linked to lethal COVID-19, and highly elevated soluble VCAM-1 and ACE2 at admission to ICU may predict unfavorable outcomes.

## 1. Introduction

During the waves of the coronavirus disease 2019 (COVID-19) pandemic, caused by the different variants of the severe acute respiratory syndrome coronavirus 2 (SARS-CoV-2), mild symptoms developed in most patients; however, 13.8% of these subjects had severe complications and 4.7% suffered from life-threatening disease with respiratory and multiorgan failure [1]. SARS-CoV-2 infection may occur at all ages under any health conditions, but patients in severe and critically ill COVID-19 status were frequently of advanced age and suffered from other underlying comorbidities, e.g., hypertension, diabetes mellitus, cardiovascular diseases, etc., causing vascular complications [2].

The endothelium normally plays a key role in the maintenance of vascular integrity and immunological responses, showing anticoagulant, antiadhesive, and vasodilatory properties [3]. In contrast, under severe inflammatory conditions, endothelial dysfunction may occur defined by the lack of one or more of these functions [4]. In COVID-19, the activation of endothelial cells in variable organs provokes the overexpression of various cytokines, chemokines and cell adhesion molecules that facilitate different cellular interactions between the endothelium and leukocytes as well as platelets [5,6], causing myocardial infarction or vasculitis and interstitial pneumonitis or acute respiratory distress syndrome (ARDS) [7,8,9]. Indirect cellular activation mechanisms are based on the extensive accumulation of pro-inflammatory and angiogenic mediators in the bloodstream, while endothelial cells are directly infected through the angiotensin-converting enzyme 2 (ACE2), a transmembrane glycoprotein with metalloprotease activity resulting in cell damage and apoptosis [10,11,12]. Due to the shedding of endothelial receptors from cell surfaces, the plasma/serum levels of these mediators become elevated, especially in severe COVID-19 [13]. However, only a limited amount of data is available on the clinical value of these biomarkers for prognosis estimation in SARS-CoV-2 infection [14].

ACE2 is mainly expressed on the surface of the nasal and bronchial epithelium and in the membrane of type II pneumocytes [15]. Coronavirus possesses a structural protein S (spike) on its surface that is composed of two subunits named S1 and S2, and this is responsible for the viral binding to host cells. The binding to the ACE2 causes a conformational change in protein S1/S2 and provokes endocytosis into epithelial cells [16]. The direct relationship between SARS-CoV-2 and ACE2 is supported by those facts that the entry of virus particles and subsequent viral replication can downregulate ACE2 expression in the host endothelial cell via ACE2 endocytosis and ectodomain shedding by the disintegrin and metalloprotease 17 (ADAM17) producing the soluble form of this receptor [17]. Hence, the concentration or activity of serum ACE2 has been thoroughly measured as a vascular biomarker in COVID-19 [18] with, however, controversial results about its level [19,20,21,22,23,24,25,26,27]. Overall, SARS-CoV-2 targets the endothelium in multiple organs, leading to endothelial injury [28], and liberated ACE2 may act as an endogenous nonspecific protective mechanism against SARS-CoV-2 infection [19].

Lipoprotein-associated phospholipase A2 (Lp-PLA2), or platelet-activating factor acetyl–hydrolase, is secreted by the macrophages and circulates in the blood in a complex with low-density lipoprotein (LDL) [29]. Lp-PLA2 hydrolyzes oxidized phospholipids to pro-inflammatory products, which are the key factors in endothelial dysfunction, plaque inflammation and the development of a necrotic core in the plaque [30]. It has been associated with the oxidative modification of LDL, resulting in an inflammatory cascade in the arterial intima. Serum Lp-PLA2 has been applied as a novel biomarker of risk stratification for the recurrence of atherosclerosis leading to coronary artery disease and stroke [31], as well as the development of a procoagulant state in lipid disorders [32]. Severe SARS-CoV-2 infection was linked to lowered levels of total cholesterol, LDL-cholesterol, and HDL-cholesterol [33] and to an increased production of oxidized lipids by injured pneumocytes and macrophages in ARDS [34]. However, only limited data are known on the biomarker potentials of Lp-PLA2 in the prediction of COVID-19-related vascular dysfunction with endothelial cell activation [35].

In this study, different vascular biomarkers were compared for predicting clinical outcome in severe SARS-CoV-2 infection. For this purpose, (i) we determined the baseline level of serum vascular adhesion molecule-1 (VCAM-1), E-selectin, ACE2 and Lp-PLA2 in parallel to platelet-specific soluble P-selectin and CD40L concentrations in severe COVID-19; (ii) the levels of these biomarkers were correlated with each other and the routinely measured prognostic markers, such as ferritin and interleukin-6 (IL-6), as well as the disease severity evaluated according to the Horowitz index; and (iii) the predictive value of these vascular dysfunction-related markers was evaluated for mortality in severe SARS-CoV-2 infection.

## 2. Materials and Methods

### 2.1. COVID-19 Patients and Clinical Controls

In this retrospective clinical study, 70 COVID-19 patients aged between 38 and 86 years were involved at the Gyula Kenézy Campus, Central Intensive Care Unit, University of Debrecen, Debrecen, Hungary between December 2020 and July 2021. At admission, these individuals suffered from life-threatening ARDS or severe pneumonia and had not received any anti-SARS-CoV-2 vaccination. All subjects were confirmed to be positive for SARS-CoV-2 by reverse transcription polymerase chain reaction (RT-PCR) test of a nasopharyngeal swab. All COVID-19 patients were then transferred to the intensive care unit (ICU). Study participants were divided into two subgroups according to the outcome of their clinical state following 30 days of hospitalization: 33 subjects were categorized as “Non-survivor COVID-19” and 37 patients were characterized as “Survivor COVID-19” (Table 1). None of these COVID-19 subjects had any bacterial infection at the time of blood sampling. All patients received the following standard treatment: oral or intravenous dexamethasone and antiviral therapy, remdesivir and erythromycin, thrombosis and gastric ulcer prophylaxis, vitamin C and D supplementation and O_2_ support or invasive ventilation, corresponding to the severity of the respiratory failure. The exclusion criteria included cancer, autoimmune disease, pregnancy, and chronic lung disease. The degree of pulmonary involvement was evaluated by chest CT, while pulmonary disease severity was defined based on the Horowitz index to assess the functional lung involvement and the impairment of organ oxygenation: PaO_2_/FiO_2_ (partial arterial pressure of oxygen (PaO_2_, mmHg)/fraction of inspired oxygen (FiO_2_) (P/F) index [36]), which was calculated by the clinicians for all patients with COVID-19. An index below 100 mmHg was defined as severe, 100–200 mmHg as moderate and 200–300 mmHg as mild respiratory failure. In parallel, 16 clinical controls with mild respiratory symptoms but negative COVID-19 RT-PCR test results were enrolled from the Department of Internal Medicine, University of Debrecen. The subjects of this control group were age- and sex-matched to both COVID-19 cohorts. The exclusion criteria for clinical controls included cancer, autoimmune disease, pregnancy, and sepsis (Table 1). The study was approved by the Scientific and Research Ethics Committee of the University of Debrecen and the Ministry of Human Capacities under the registration number of 32568-3/2020/EÜIG.

### 2.2. Laboratory Analyses

All study participants had baseline peripheral venous blood samples drawn into Vacutainer^®^ tubes within the first 24 h of hospital admission to avoid the modulating effects of SARS-CoV-2-related treatment. Routinely available hematology and chemistry tests were performed at the Department of Laboratory Medicine, University of Debrecen. Sera and plasma specimens were stored at −70 °C; analyses of serum VCAM-1 and E-selectin, as well as plasma P-selectin and CD40L, were retrospectively performed using commercially available ELISA kits based on the manufacturer’s protocol (R&D Systems, Minneapolis, MN, USA). In parallel, Lp-PLA2 was analyzed via an automated immunoassay on a Snibe Maglumi^®^ 800 instrument according to the manufacturer’s instructions (Snibe, Shenzhen, China). The analysis of serum ACE2 activity was performed by a specific quenched fluorescent substrate (obtained from http://peptide2.com (accessed on 16 September 2022) as reported earlier [18,19,20]. Routinely available laboratory serum tests, i.e., C-reactive-protein (CRP), procalcitonin (PCT), IL-6 and ferritin, were determined via electro-chemiluminescent immunoassays on a Cobas^®^ e411 analyzer (Roche Diagnostics, Mannheim, Germany), while enzyme activities (i.e., AST, ALT, LDH) and creatinine with urea levels were analyzed via kinetic colorimetric assays on a Cobas^®^ 8000 instrument (Roche Diagnostics). Finally, hematology parameters, i.e., white blood cell (WBC) count, platelet (PLT) count and mean platelet volume (MPV) values, were determined using an Advia^®^ 2120 Hematology System analyzer (Bayer Diagnostics, Tarrytown, NJ, USA).

### 2.3. Statistical Analysis

Before the enrollment of study participants, a prior sample size calculation was performed using the Intercooled Stata v17 with a power level of 90% and an α level of 0.05 using the median (33.7 and 60.6 ng/mL) and IQR (25.0–45.5 and 49.2–111.6 ng/mL) of serum E-selectin measured in survivor and non-survivor COVID-19 ICU cohorts based on a previous study [14]. Accordingly, the estimated required sample sizes were 33 patients per study group. The Shapiro–Wilk test was used for evaluating the normality of data. Results are expressed as median with interquartile range (IQR). To compare the data of two groups, we applied the Mann–Whitney U test or Fisher’s exact test, as appropriate. A comparison of multiple groups was performed by Kruskal–Wallis test with Dunn’s multiple comparisons test. Correlations between the levels of biomarkers and other clinical and laboratory parameters were determined using Spearman’s test. The area under the receiver operating characteristic curve (ROC-AUC) value was determined for the baseline concentration of the measured parameters to indicate the clinical outcome of severe COVID-19 disease. The maximum Youden index was determined to identify the cut-off values. Statistical significance was defined when *p* value was < 0.05. Analyses were performed using GraphPad Prism, version 9 (GraphPad Software, La Jolla, CA, USA).

## 3. Results

### 3.1. Baseline Characteristics of COVID-19 Patients and Clinical Controls

In total, 70 severe COVID-19 individuals were involved in this study, who were subgrouped according to the clinical outcome (Table 1). Regarding age and sex, there was no difference among these study subgroups. The length of hospital stay was similar between non-survivors and survivors (median (IQR), 9.0 (5.5–16) vs. 10 (6.5–13) days). Mechanical ventilation was more frequently applied among those severe COVID-19 patients who died of ARDS vs. survivors (30 out of 33 vs. 2 out of 37 individuals, *p* < 0.0001). In addition, 16 age- and sex-matched non-COVID-19 clinical controls showed significantly lower levels in various laboratory parameters (e.g., CRP, ferritin, LDH) compared to non-survivors and convalescent COVID-19 cases as well. General inflammatory markers, such as CRP, PCT, IL-6, ferritin, and white blood count (WBC), were significantly higher (*p* < 0.001 or *p* < 0.0001, respectively) among COVID-19 non-survivors compared to survivors, while PLT count and MPV did not show a significant difference between the two subgroups (Table 1). The Horowitz index was calculated in the cases of all patients suffering from COVID-19 and was significantly lower in the cohort of non-survivors vs. survivor patients (95 (65–157) vs. 196 (138–381), *p* < 0.001). Furthermore, lung manifestations were further evaluated via chest CT images, and deceased subjects suffered from more advanced pulmonary involvement compared to convalescent patients (73 (60–80) vs. 50 (30–70) %, *p* < 0.05), which highly contributed to the overall disease outcome. Importantly, no significant difference was observed in pre-COVID-19 comorbidities between the two COVID-19 subcohorts (Table 1).

### 3.2. Increased Baseline VCAM-1 and ACE2 Serum Levels Highly Reflect the Degree of Vascular Dysfunction in COVID-19 Progression

First, baseline serum levels of VCAM-1, E-selectin, ACE2 and Lp-PLA2 were retrospectively measured to evaluate the degree of vascular disorders with endothelial cell activation and to compare them among study sub-cohorts (Figure 1A–D). Both the soluble VCAM-1 concentration and ACE2 activity showed a statistically significant elevation (*p* = 0.0001 and *p* = 0.0038, respectively) at baseline in non-survivors vs. controls (Figure 1A,C). Moreover, both parameters were significantly altered regarding overall survival. In contrast, E-selectin levels demonstrated only a “borderline” significance (*p* = 0.0656) between deceased cases and controls (Figure 1B), while Lp-PLA2 did not alter in COVID-19 (Figure 1D). In parallel, plasma P-selectin and CD40L levels were analyzed as platelet-activation-dependent markers and compared to controls; soluble P-selectin was significantly higher (*p* = 0.0072) in non-survivors (Figure 1E), while CD40L values were significantly augmented in both COVID-19 subgroups (*p* = 0.0012 and *p* = 0.0022, respectively), but no difference was observed between survivors and non-survivors (Figure 1F). Based on these data, VCAM-1 concentration and ACE2 activity at ICU admission effectively reflected the degree of endothelial dysfunction.

Next, we statistically investigated the relationship among the selected vascular biomarkers of age, Horowitz index and the routinely determined laboratory parameters using Spearman’s test. Age did not alter the level of any of these parameters; however, we excluded its potential influence on current tests via recruiting age-matched patient populations. Importantly, a significant correlation was found between VCAM-1 and the Horowitz index (r = 0.3115), PCT (r = 0.3664), IL-6 (r = 0.4599) and E-selectin (r = 0.3643), while an inverse association was analyzed with renal dysfunction evaluated by GFR (r = −0.5076) (Table 2). We also examined the correlation between serum E-selectin/ACE2 and inflammation-dependent parameters, as well as some prognostic markers (i.e., total LDH activity and ferritin). Our data confirmed that both endothelial biomarkers were significantly correlated with CRP, PCT and IL-6 levels, in addition to LDH activity and ferritin. Surprisingly, serum Lp-PLA2 did not show any relationship with other parameters. When platelet activation markers (i.e., soluble P-selectin and CD40L) were studied for correlations with the endothelial parameters, a moderate but significant link was shown between ACE2 and CD40L (r = 0.2948) (Table 2). These results suggest that, among these biomarkers above, serum VCAM-1 concentrations and ACE2 activity were highly modulated by COVID-19 clinical status in response to vascular inflammation and pulmonary dysfunction, while enhanced platelet reactivity and lipid peroxidation did not substantially affect these vascular markers in our subjects.

### 3.3. Efficacy of Baseline Serum VCAM-1, E-Selectin and ACE2 Levels for Early Indication of Unfavorable Disease Outcome in Severe COVID-19

To further evaluate the clinical usefulness of these serum biomarkers, which showed considerable changes (as described above), as new prognostic biomarkers in COVID-19, we statistically analyzed their diagnostic characteristics to predict the disease outcome using ROC-AUC curve analyses (Figure 2A–C). The best discriminative threshold of VCAM-1 levels at admission, estimated according to the Youden index, was 1420 ng/mL with a sensitivity of 64% and a specificity of 73% to estimate disease outcome at an AUC value of 0.6855 (95% CI [0.5595–0.8115], *p* = 0.0077) (Figure 2A). In the case of serum E-selectin concentration and ACE2 activity, very similar AUC values were determined to distinguish non-survivors from survivors. The ideal cut-off value of baseline E-selectin was 49.3 ng/mL with a sensitivity of 58% and a specificity of 70.3% to predict the outcome of COVID-19 with an AUC value of 0.6523 (95% CI [0.5215–0.7813], *p* = 0.0286) (Figure 2B). Finally, serum ACE2 activity had an AUC value of 0.6519 (95% CI [0.5213–0.7826], *p* = 0.0291) with a 57.6% sensitivity and 70.4% specificity at the cut-off value of 45.2 μU/mL (Figure 2C). Based on these results, baseline VCAM-1, E-selectin and ACE2 demonstrated equal effectiveness in assessing the progression of severe COVID-19 disease based on ROC-AUC analysis.

### 3.4. Prediction of 30-Day Mortality by Elevated Baseline VCAM-1 and ACE2 in Patients with Severe COVID-19

Out of the 70 recruited COVID-19 patients, 33 died during the 30-day follow-up. As we stated before, VCAM-1, E-selectin and ACE2 efficiently estimated the outcome of this disease based on the ROC-AUC curve analysis. Using those cut-off values in the Kaplan–Meier analysis, COVID-19 patients with highly elevated VCAM-1 levels had a significantly higher risk of 30-day mortality compared to those with lower values (with a death ratio of 68% vs. 37%, respectively, Log rank *p* = 0.0031) (Figure 2D). Additionally, the baseline ACE2 activity of ≥45.2 μU/mL was linked to a higher risk of death with a mortality ratio of 65.7% vs. 40%, respectively, Log rank *p* = 0.0117) (Figure 2F). In contrast, increased levels of E-selectin prior to treatment (≥49.3 ng/mL) almost significantly (*p* = 0.0707) predicted mortality in severe COVID-19 (Figure 2E). Taken together, serum VCAM-1 and ACE2 at ICU admission had a capacity to predict mortality in COVID-19.

## 4. Discussion

Endothelial dysfunction plays a prominent role in the pathomechanism and the progression of COVID-19 disease [6,8,37]. As a result of recent extensive research, abundant information has now become available about the changes in various cytokines and adhesion molecules during hyper-inflammation of COVID-19 (Figure 3) [38]. Interestingly, there has been a debate whether endothelial cells can be directly infected by SARS-CoV-2 via ACE2 or not. This phenomenon was supported by those data when virus particles were detected in lung microvascular endothelial cells [39], in kidney endothelial cells [40], in brain capillary endothelial cells [41] and in the capillary endothelial cells of the skin [42] obtained from severe COVID-19 individuals. In contrast, in other autopsy samples analyzed by immunohistochemistry or electron microscopy, no positivity for SARS-CoV-2 virus was found in pulmonary endothelial cells [43,44].

Once endothelial cells become activated by SARS-CoV-2, their antithrombotic phenotype is disturbed and prothrombotic features appear, showing an increased surface expression of several adhesion receptors, such as E-selectin, VCAM-1, intercellular adhesion molecule 1 (ICAM-1), platelet endothelial cell adhesion molecule 1 (PECAM-1), vascular endothelial growth factor (VEGF), ephrin-A1 and ephrin type-A receptor 2, soluble triggering receptor expressed on myeloid cells 1 (sTREM-1) and downregulated anticoagulant properties, e.g., thrombomodulin and heparan sulfate [45,46,47]. In parallel, platelets also become activated under this hyper-inflammatory milieu, which participate in the formation of heterotypic aggregates with leukocytes and platelet–endothelial cell interactions [48]. Consequently, there is a higher risk for coagulopathy and thromboembolism contributing to increased mortality in COVID-19 [49]. To date, only a few biomarkers are applied in clinical practice to predict the outcome of severe SARS-CoV-2 infection, as reviewed in [50]. Hence, the aim of this current study was to further analyze the level of some soluble vasculature-linked molecules and to correlate them with routine parameters, with platelet-activation-dependent molecules as well as with COVID-19 disease outcome.

The surface and cytosol of endothelial cells demonstrate several types of ancient pattern recognition receptors, such as toll-like receptors (TLRs) [4]. As a part of the indirect endothelial activation by SARS-CoV-2, the activation of TLRs by different cytokines also induces a hyperinflammatory response via the intracellular NF-κB and MAPK pathways, resulting in the additional production of IL-6, IL-8, IL-1β and adhesion molecules (ICAM-1, E-selectin, etc.) [37]. Early investigations reported much higher antigen levels and von Willebrand factor (vWF) activity between ICU and non-ICU COVID-19 patients, which were inversely associated with PaO_2_/FiO_2_ values [51]. In parallel, increased serum levels of VCAM-1 were described in COVID-19 patients with distinct severity under dexamethasone treatment in correlation with classic sepsis biomarkers soluble urokinase-type plasminogen activator receptor and presepsin [52]. Recently, Lázaro et al. proved that an even more augmented concentration of VCAM-1, with a high acute-lung-injury-specific angiopoeitin-2/angiopoietin-1 ratio, predicted unfavourable outcomes with a good efficacy in critically ill COVID-19 patients with diabetes mellitus [53]. In our current study, serum VCAM-1 showed a significant elevation in both survivors and non-survivors vs. controls and effectively distinguished these two COVID-19 subgroups from each other. This biomarker had a good predictive value for poor prognosis, regardless of the comorbidities of our COVID-19 patients, similar to previous findings that found an even larger AUC value of 0.91 [54].

Likewise, soluble E-selectin was analyzed as a sensitive endothelium marker, which demonstrated controversial results in variable COVID-19 cohorts. Most studies agreed that E-selectin concentration at hospital admission became elevated, suggesting the development of endothelial dysfunction [14,54,55]; however, Gelzo et al. did not find any difference between asymptomatic and hospitalized COVID-19 patients [56]. On the other hand, soluble E-selectin was significantly correlated with neutrophil count and the number of days from symptom onset to hospitalization [54]. Furthermore, increased plasma E-selectin levels (≥32.5 ng/mL), along with P-selectin and L-selectin, were independent predictors for thrombosis in hospitalized COVID-19 subjects [55]. When ICU survival probability and mortality were statistically analyzed in critically ill COVID-19 individuals, soluble E-selectin measured at ICU admission could predict the time to death in high-risk and low-risk subgroups and showed a remarkable AUC value of 0.88 with 100% sensitivity and 75% specificity [14]. Here, we found that serum E-selectin had a “borderline” significant increment in non-survivors compared to controls. In terms of the prediction of disease outcome, although E-selectin had a significant AUC value of 0.6523, using the cut-off value of 49.3 ng/mL, no significant difference was found in this study using Kaplan–Meier curve analysis.

In the last three years, ACE2 has come into focus in SARS-CoV-2 research, as this receptor has a functional role in the direct infection of endothelial cells by SARS-CoV-2 [18,28]. Surprisingly, controversial results have been published about the level of soluble ACE2; elevated activity values with a strong correlation with disease severity and outcome were described in many clinical studies [19,20,21,22,23,24], while unchanged levels [25,26] or even lower quantities [27] were explored in groups of severe COVID-19 patients. Here, we confirmed our previous data [19,20] by showing that ACE2 activity was increased in the acute phase of severe SARS-CoV-2 infection and predicted mortality in hospitalized COVID-19 patients. There was a significant difference in ACE2 levels between the two sub-cohorts with distinct outcomes, showing a significant AUC value (0.6519) as well as Log-Rank *p* value (*p* = 0.0117) for the prediction of disease progression in severe COVID-19 conditions. This biomarker was also correlated with E-selectin, CD40L and inflammation-dependent routine parameters (i.e., CRP, PCT, IL-6).

Lp-PLA2 can be useful as an atherosclerosis-specific biomarker for risk stratification in coronary artery disease and stroke [31], as well as lipid disorders [32]. The normal serum level of Lp-PLA2 is less than 200 ng/mL [29]. Altered levels of different forms of cholesterol were detected in severe SARS-CoV-2 infection [33] and increased amounts of oxidized lipids were reported in COVID-19-related ARDS [34]. Hence, we decided to analyze the level of Lp-PLA2 as a prognostic marker for disease outcome, because only limited data are available on Lp-PLA2 as a predictor of COVID-19-related death [35]. The serum protein level of Lp-PLA2 (or PLA2G7) was found to be elevated in COVID-19 subjects and its expression was highly induced in macrophages in the lung [57]. Furthermore, Carmo et al. investigated the longitudinal changes in serum Lp-PLA2 levels in hospitalized COVID-19 patients and found a progressive increase during the first 30 days, regardless of the outcome [58]. In our COVID-19 subjects, we did not detect any differences in Lp-PLA2 analyzed upon hospital admission in either survivors or deceased patients. Surprisingly, baseline Lp-PLA2 did not show any correlation with classic inflammatory factors and was not associated with disease progression, thus we could not underline its predictive potential in COVID-19, as has been suggested by others [35,57,58]. The reason for this discrepancy may be due to the distinct clinical features of the studied COVID-19 populations. Since this marker is more associated with atherosclerosis [31], and there was no difference in the presence of coronary artery disease between our two COVID-19 cohorts (Table 1), this is why we think that severe SARS-CoV-2 infection could not induce Lp-PLA2 activity in these patients.

Since there is a direct link between endothelial and platelet activation in COVID-19 [48], we described the level of platelet activation in these subjects via plasma P-selectin and CD40L levels and correlated them with endothelial dysfunction parameters. According to a previous study, the elevation of soluble P-selectin had a strong relationship with the incidence of thromboembolic events and the probability of unfavorable outcome in COVID-19 disease [55]. Here, we measured a significantly higher concentration of P-selectin in cases of critically ill non-survivors compared to the control group, while severe COVID-19 survivors had only modest elevations. In contrast, CD40L plasma levels were already augmented in convalescent patients, not only in deceased individuals, and were significantly correlated with ACE2 activity. CD40L is a multifunctional molecule produced by T-cells and platelets that is related to the intercellular communication of the adaptive immune response [59]. Already at the early moderate stages of COVID-19, induced levels of soluble CD40L with increased plasma P-selectin, fibrinogen and plasminogen activator inhibitor-1 indicated the activation of platelets and coagulation system upon SARS-CoV-2 infection [60]. However, additional parameters, such as soluble ICAM-1, could also be effective in evaluating endothelial dysfunction via significantly elevated levels in non-survivors compared to survivors and via a positive correlation with fibrinogen [14]. Notably, plasma ICAM-1 was related to a higher risk of death, showing an AUC value of 0.86 with a 90% sensitivity and 71.4% specificity [14]. Recently, persistent endotheliopathy, marked by elevated levels of plasma vWF or vWF/ADAMTS13 ratio, was present in all hospitalized patients following severe SARS-CoV-2 infection, which was strongly associated with mortality [61].

In summary, enhanced baseline levels of VCAM-1 and ACE2 can help to identify those severe COVID-19 patients who may subsequently die in ICUs (Figure 3). However, it is still debated whether these biomarkers demonstrate vascular injury or merely endothelial cell activation. Either way, if there is a higher chance of a substantial endotheliopathy with an increased need for O_2_ supply under such COVID-19 conditions, this requires comprehensive clinical management, including the administration of more than one immunomodulatory drug, as well as anticoagulation and anti-platelet medications [62]. We think that these biomarkers may facilitate the closer monitoring of critically ill patients via more frequent blood gas analysis, blood pressure measurement, chest CT examination, laboratory tests, etc.

This study has some limitations. First, this was a single-center study including a moderate number of relatively older COVID-19 patients; however, these cohorts are well characterized and comparable to similar previous studies conducted during the same period of the COVID-19 pandemic. Regarding the statistical analysis, multivariate logistic regression analysis was not performed due to the limited number of patients. The recruitment of patients into sub-groups was set according to the opinion of clinicians and clinical data; thus, the cut-off values obtained from the ROC curves were used to examine the cumulative effects of the biomarkers on mortality. A cross-center evaluation with a larger sample might provide more accurate results. Second, we analyzed only one blood sample taken at hospital admission which serves as a “snapshot” of the acute phase of vascular functions; however, the expression of these investigated molecules may further alter under treatment and in disease progression. Third, these patients involved had not received any vaccination; thus, we could not evaluate the potential effect of anti-SARS-CoV-2 vaccines on the expression of vascular biomarkers.

## 5. Conclusions

Substantial endothelial dysfunction is linked to mortality in severe COVID-19 states. Among the several biomarkers, the soluble VCAM-1 and ACE2 levels determined at ICU admission showed the most considerable elevation and diagnostic power; they may successfully predict adverse clinical outcome upon severe SARS-CoV-2 infection.

## Figures and Tables

**Figure 1 microorganisms-12-00229-f001:**
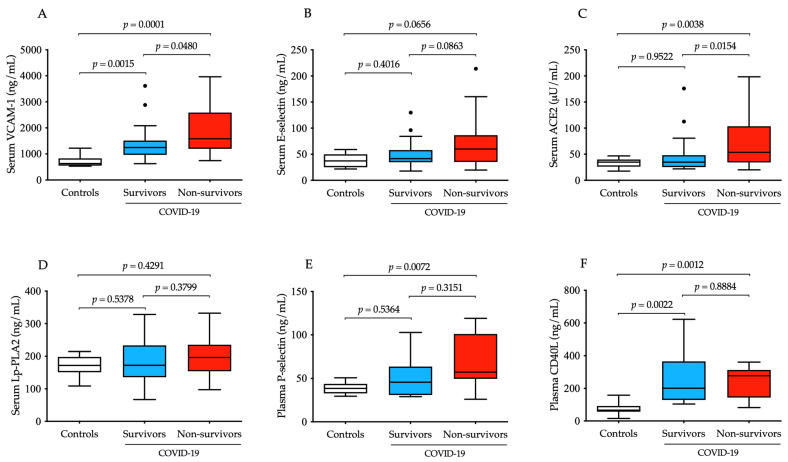
Comparison of baseline serum concentration of four different vascular dysfunction parameters in parallel with two platelet-specific biomarkers in severe COVID-19 patients with distinct disease progression in contrast to non-COVID-19 clinical controls. Significantly higher baseline levels of VCAM-1 were measured in survivors (n = 37, blue) and non-survivors (n = 33, red) compared controls, and there were significant differences among two COVID-19 sub-cohorts (**A**). There were relatively higher pre-treatment E-selectin levels in non-survivors vs. controls; however, this change did not reach significance (**B**). ACE2 activity before treatment was elevated in deceased cases and was also significantly higher than in survivors (**C**). Serum Lp-PLA2 showed no tendency in COVID-19 (**D**). Platelet-activation-dependent biomarkers (soluble P-selectin and CD40L) had significantly augmented concentrations in non-survivors but could not distinguish the two COVID-19 sub-cohorts from each other (**E**,**F**). To compare the data of these groups, the Kruskal–Wallis test with Dunn’s multiple comparisons test was applied. Median and IQR are depicted and values out of range are indicated with dots.

**Figure 2 microorganisms-12-00229-f002:**
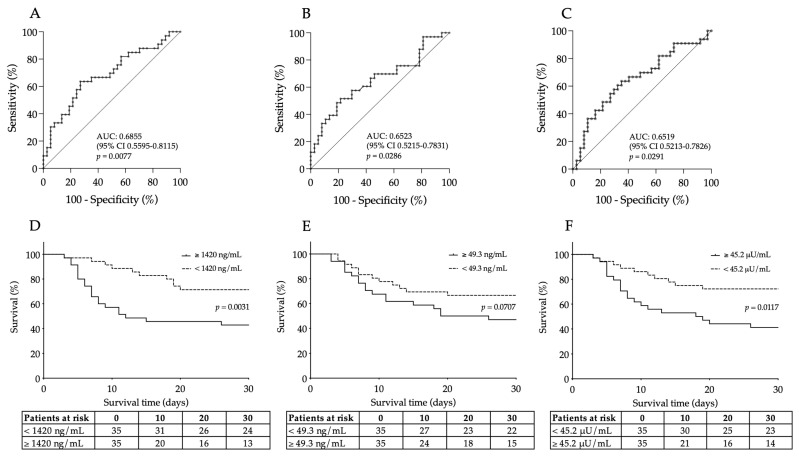
ROC curve analysis was performed for baseline VCAM-1 (**A**), E-selectin (**B**) and ACE2 levels (**C**) for the prediction of unfavorable outcomes of COVID-19. The best discriminative cut-off values of all these serum biomarkers were determined to estimate disease progression with their ROC-AUC values. Similar efficacy was observed for these endothelial biomarkers. Kaplan–Meier analysis indicated that highly elevated VCAM-1 (**D**) and ACE2 levels (**F**) were significantly related to a higher rate of 30-day mortality based on those cut-off values set by ROC-AUC tests, while a “borderline” *p* value was shown for E-selectin (**E**). Number of patients at risk are displayed at given days and Log rank *p* value was determined for each parameter.

**Figure 3 microorganisms-12-00229-f003:**
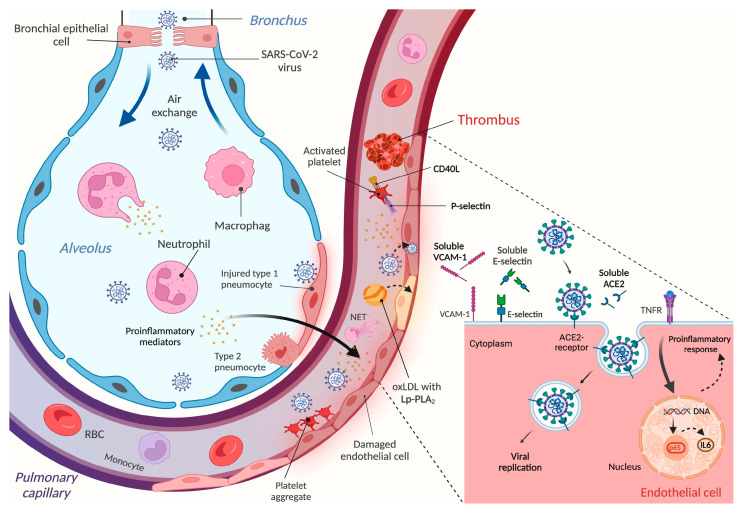
Schematic figure on the mechanism of endothelial cell activation and vascular dysfunction upon SARS-CoV-2 infection. After virus particles enter the alveolus, leukocytes become activated and the expression of proinflammatory cytokines is enhanced, interfering with normal pulmonary functions. SARS-CoV-2 and various mediators can then enter the circulation via lung capillaries and elicit hyper-inflammation responses that stimulate different cell types, such as platelets, neutrophils, and endothelial cells. Enhanced lipid peroxidation may also contribute to endothelial disorders. Activated platelets can bind to injured endothelium and participate in thrombus formation; neutrophils can induce NETosis, while endothelial cells become dysfunctional and facilitate the development of thromboembolism. This latter complication may cause ischemic failure and related comorbidities in several organs. Due to the shedding mechanism of variable endothelium surface receptors, the soluble form of these proteins can accumulate in the plasma/serum, representing additional biomarkers for reflecting endothelial malfunctions. This figure was designed by Biorender.com.

**Table 1 microorganisms-12-00229-t001:** Baseline demographical, clinical, and routine laboratory parameters of 70 COVID-19 patients divided into two subgroups based on the outcome of their disease, along with 16 non-COVID-19 clinical controls.

Variables	Non-Survivor COVID-19 (n = 33)	Survivor COVID-19 (n = 37)	Clinical Controls (n = 16)
Age (years) (median, IQR)	65 (55–74)	61 (52–65)	63 (54–72)
Sex (M/F)	20/13	20/17	9/7
Hospital stay (days) (median, IQR)	9.0 (5.5–16)	10 (6.5–13)	-
Mechanical ventilation (y/n, %)	30/3 (91) ***	2/35 (6)	-
Horowitz index (P/F) (median, IQR)	95 (63–157) **	196 (138–381)	-
Lung manifestation by CT (%) (median, IQR)	73 (60–80) *	50 (30–70)	-
CRP (mg/L)	184 (135–283) **	107 (29–237)	2.5 (0.9–9.8) ^‡‡‡^
PCT (μg/L)	0.7 (0.3–8.8) ***	0.05 (0.05–0.27)	n.d.
IL-6 (ng/L)	101.1 (43.2–206.3) ***	26.7 (3.6–51.0)	n.d.
Ferritin (μg/L)	1144 (777–2455) **	596 (294–1122)	167 (70–268) ^‡‡‡^
Urea (mmol/L)	8.7 (6.7–15.2) *	5.4 (4.4–6.9)	5.6 (4.9–6.0)
Creatinine (μmol/L)	108.0 (81.0–183.0) *	88.0 (77.5–100.0)	64.0 (59.0–78.0) ^‡^
GFR CKD-EPI (mL/min/1.73 m^2^)	79 (36–79) *	65 (59–76)	89.0 (75.2–90.0) ^‡^
AST (U/L)	46.0 (37.5–55.0)	36.0 (26.5–49.5)	19.5 (15.0–25.3) ^‡^
ALT (U/L)	36.0 (23.0–66.0)	36.0 (25.5–52.0)	22.0 (14.5–29.8) ^‡^
LDH (U/L)	770 (671–1083) **	498 (355–725)	192 (179–244) ^‡‡^
WBC count (G/L)	9.9 (6.2–13.8) *	7.6 (5.2–9.8)	7.8 (5.9–9.5)
PLT count (G/L)	235 (156–292)	205 (145–255)	249 (177–307)
MPV (fL)	8.3 (7.5–9.9)	8.1 (7.2–8.9)	8.2 (7.8–9.8)
Hypertension (y/n, %)	27/6 (82)	28/9 (76)	10/6 (63)
Diabetes mellitus	10/23 (30)	11/26 (29)	3/13 (19)
Coronary artery disease	14/19 (42)	15/22 (41)	2/14 (13) ^‡^
Atrial fibrillation	12/21 (36) *	4/33 (11)	0/0 (0) ^‡^
Renal insufficiency	13/20 (39) *	3/34 (8)	0/0 (0) ^‡^

Data are expressed as median with IQR. For statistical analyses, Mann–Whitney U test or Fisher’s exact test was used, as appropriate. Significant differences between non-survivors vs. survivor COVID-19 patients were found in different comparisons, indicated with variable symbols as follows: *** *p* < 0.0001, ** *p* < 0.001, * *p* < 0.05; and ^‡‡‡^
*p* < 0.0001, ^‡‡^
*p* < 0.001, ^‡^
*p* < 0.05 upon comparison between convalescent COVID-19 patients and controls. Abbreviations: CRP: C-reactive protein, PCT: procalcitonin, IL-6: interleukin-6, GFR-CKD-EPI: Glomerular filtration rate—Chronic Kidney Disease Epidemiology Collaboration, AST: aspartate transaminase, ALT: alanine aminotransferase, LDH: lactate dehydrogenase, WBC: white blood cell, PLT: platelet, MPV: mean platelet volume, n.d.: not determined.

**Table 2 microorganisms-12-00229-t002:** Correlations between baseline VCAM-1, E-selectin, ACE2 and Lp-PLA2 levels and some important clinical and routine laboratory parameters using Spearman’s tests.

	VCAM-1	E-Selectin	ACE2	Lp-PLA2
**Age**	r = 0.1290	r = 0.1589	r = 0.1497	r = 0.1865
**Horowitz index**	**r = 0.3115 ***	r = 0.1251	r = 0.0602	r = 0.3551
**CRP**	r = 0.2991	**r = 0.2885 ***	**r = 0.4602 *****	r = 0.2003
**PCT**	**r = 0.3664 ****	**r = 0.3336 ****	**r = 0.3484 ****	r = 0.1896
**IL-6**	**r = 0.4599 *****	**r = 0.3649 ****	**r = 0.2373 ***	r = 0.1232
**Ferritin**	**r = 0.3209 ****	**r = 0.5220 *****	**r = 0.4461 *****	r = 0.2396
**Total LDH**	r = 0.1471	**r = 0.2995 ***	**r = 0.4297 ****	r = 0.1479
**GFR-CKD-EPI**	**r = −0.5076 *****	r = −0.2223	r = −0.2498	r = −0.1414
**WBC count**	r = 0.2154	**r = 0.2735 ***	r = 0.1598	r = 0.1899
**E-selectin**	**r = 0.3643 ****	r = 1.0000	**r = 0.4143 *****	r = 0.1998
**P-selectin**	r = 0.1834	r = 0.1781	r = 0.0954	r = 0.1500
**CD40L**	r = 0.3182	r = 0.1889	**r = 0.2948 ***	r = 0.1190

VCAM-1 showed a significant correlation with the Horowitz index, while soluble E-selectin and ACE2 were also associated with inflammation-specific biomarkers, such as PCT, IL-6, ferritin, and E-selectin. Lp-PLA2 had no direct relationship with any of these tested parameters. Statistically significant relationships were marked with bold letters. * *p* < 0.05, ** *p* < 0.001, *** *p* < 0.0001. Abbreviations: CRP: C-reactive protein, PCT: procalcitonin, IL-6: interleukin-6, LDH: lactate dehydrogenase, GFR-CKD-EPI: Glomerular filtration rate—Chronic Kidney Disease Epidemiology Collaboration, ACE2: Angiotensin-converting enzyme 2, WBC: white blood cell.

## Data Availability

All data are contained within the article.

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
