# Peer review of "Comparison of Different Vascular Biomarkers for Predicting In-Hospital Mortality in Severe SARS-CoV-2 Infection"

_microorganisms, 2024, doi:10.3390/microorganisms12010229_

Round 1
Reviewer 1 Report
Comments and Suggestions for Authors
Adjust the abstract to the standardized word count for the journal
The introduction is extremely long
The sample of 70 patients is tiny. I don't know how representative this sample size is. What sample size do similar studies have?
How would these biomarkers behave after vaccination? Since your sample is a predecessor of vaccination in December 2020-July 2021
Is endothelial activation comparable to viral load?
Is endothelial activation not related to the presence of comorbidities in patients? More clinical data from patients is needed.
Would these biomarkers only apply to elderly patients? Their groups are between 52 and 74 years old.
Comments on the Quality of English LanguageMinor editing of English language required
Reviewer 2 Report
Comments and Suggestions for Authors
In this manuscript, Renata Suto et al. sought to determine the degree of endothelial dysfunction in severe SARS-CoV-2 infection and whether it can serve as prognostic indicator in the disease progression. The authors collected serum from 70 severe COVID-19 patients (survivors and non-survivors) and evaluated the levels of a group of various markers that may or may not reflect the endothelial functions as well as other inflammatory molecules/cytokines including Lp-PLA2, CRP, IL-6, Ferritin et c.. In addition, serum ACE2 activity was also measured as cardiovascular biomarker. The results of this study showed that only VCAM-1 and ACE2 activity can be considered as makers to predict the prognosis or severity of COVID -19. Although the goal of the study is to retrospectively study the patient specimens to discover useful prognostic biomarkers, which should be highly significant, the manuscript suffered from major defects, notably inadequate background analysis and unclear data presentation. Furthermore, while the study intended to identify useful prognostic biomarkers, it failed to demonstrate a strong correlation between the measured molecules, endothelial function markers, and patient survival. Therefore, the main problem is that the conclusion of the study was not supported by the results.
Major comments:
1) It remains unclear whether the authors intended to investigate biomarkers for disease severity or patient survival in severe COVID-19. The manuscript appears to use these terms interchangeably, lacking clarity in distinguishing between them.
2) An accurate review of all the molecules serving as indicators for endothelial function in COVID-19 was crucial for this study's scope.
3) Several “above mentioned “appeared in the manuscript. It was not clear to me which is which.
4) Insufficient interpretations and analyses were provided when the results contradicted previous findings in the literature. For instance, if IL-6 and Ferritin were previously considered as biomarkers for COVID-19, the discrepancies in the results were not adequately addressed.
5) All the molecules measured in the study seem not correlated with one focused hypothesis.
6) The interpretations of results regarding CD40L were unclear and less pertinent to endothelial dysfunction.
7) The presentation of data throughout the manuscript could benefit from improvement.
Minor comments:
1) It is highly recommended that the manuscript needs English edits. For example, Page on Line 28, it was not clear why “either” as used here…; Page 3 Line 112, it is confusing.
2) The abbreviations were introduced in the article without initial explanation, such as Lp-PLA2.

Comments on the Quality of English LanguageThe manuscript will benefit from English edits.
Reviewer 3 Report
Comments and Suggestions for Authors
Since the work is now dated having been conducted in 2020-2021 and the standards for care were different than now, I would put this factor among the limitations.
I would add a paragraph explaining well how this clinical data of the serum dosage finding of V CAM-1 and ACE-2 can help the clinician in the management of this disease.
Also, it is important to specify whether the patients had been vaccinated or not, whether vaccines were not yet widespread at that time in Hungary, or whether being vaccinated was a protective factor, also in light of the serum assays of these biomarkers.
A paper that conducted now would be fantastic, but one that is affected by these factors, you really will have to give it your all to make it current and useful to scholars reading it now, at this time, when treatments and mortality for covid-19 have changed dramatically and when the fact that immunization has occurred because of the vaccine has really changed everything.
I expect substantial changes.
Round 2
Reviewer 3 Report
Comments and Suggestions for Authors
the authors have modified what was requested, that's fine